# Clinical Phenotypes of Severe Cow’s Milk Protein Allergy with Various Responses to Amino Acid-Based Formula

**DOI:** 10.3390/nu17111809

**Published:** 2025-05-26

**Authors:** Łukasz Błażowski, Daniela Podlecka, Agnieszka Brzozowska, Joanna Jerzyńska, Michał Seweryn, Marcin Błażowski, Paweł Majak

**Affiliations:** 1Department of Allergology and Pulmonology, National Research Institute of Tuberculosis and Lung Diseases, 34-700 Rabka-Zdroj, Poland; l.blazowski@gmail.com; 2Department of Pathophysiology, Institute of Medical Sciences of Rzeszow University, 35-310 Rzeszow, Poland; 3Department of Pediatrics and Allergy, Medical University of Lodz, 90-419 Lodz, Poland; agnieszka.brzozowska@umed.lodz.pl (A.B.); joanna.jerzynska@umed.lodz.pl (J.J.); 4Korczak Pediatric Center, 92-328 Lodz, Poland; pawel.majak@umed.lodz.pl; 5Biobank Laboratory, Department of Molecular Biophysics, University of Lodz, 90-136 Lodz, Poland; michal.seweryn@biol.uni.lodz.pl; 6Specialist Hospital, 38-200 Jaslo, Poland; m.blazowski@gmail.com; 7Department of Pediatric Pulmonology, Medical University of Lodz, 90-419 Lodz, Poland

**Keywords:** cow’s milk allergy, infants, children, cluster analysis, phenotypes, treatment, amino acid-based formula

## Abstract

Background: The symptoms of cow’s milk allergy (CMA) can vary widely in severity and course, so diagnosis and treatment are still challenging. Objective: This study aims to establish the phenotype of severe CMA in children with the greatest improvement following the application of amino acid-based formula (AAF). Methods: This is a post hoc analysis of data from the multicenter, real-life study assessing the clinical effectiveness of a 5-week AAF intervention in 232 infants with severe CMA. A cluster analysis based on symptom severity at the 1st visit was performed. The differences in the severity scale of each symptom before and after the intervention were assessed and compared within and between clusters. The clustering results were validated in a separate cohort of infants with CMA (n = 157). Results: Three clusters were identified: cluster A (38.8% of patients) with moderate-to-severe gastrointestinal symptoms, cluster B (34.1%) with severe skin symptoms, and cluster C (25.9%) with combined moderate-to-severe gastrointestinal and severe skin symptoms. In the validation cohort, three clusters with the same pattern of symptoms were observed among children with moderate-to-severe CMA. The multivariate model of linear regression showed that severity score reductions after AAF treatment were significantly higher in cluster C than in clusters A and B, in children with a positive family history of allergy, and in children with growth retardation at baseline. Conclusion: Symptoms of severe CMA in children are grouped into three distinct phenotypes—gastrointestinal, skin, and combined gastrointestinal and skin. The most significant improvement after AAF implementation was obtained in patients with a combined phenotype.

## 1. Introduction

In recent decades, the incidence of food allergy and food anaphylaxis has increased significantly [1,2,3]. In the pediatric population, especially in children up to 2 years of age, the most common cause of food allergy is cow’s milk protein allergy (CMA), with a global prevalence of 1.9–4.9% [4,5]. The diagnosis of CMA in many children remains a challenge despite the relatively high diagnostic properties of available tools [6]. According to the reports of many researchers, it is currently believed that the early manifestation of CMA is the first link of the allergic march, and in the future, it may lead to the development of atopic dermatitis, allergic rhinitis, and asthma [7,8,9].

The immune reaction to cow’s milk proteins can be IgE-mediated, developing within minutes to 2 h, manifesting as a delayed non-IgE-mediated or mixed IgE and non-IgE mediated reaction [10]. Since the symptoms of CMA in small children can vary widely in form, severity, and course, both diagnosis and the choice of treatment remain challenging [7,8]. The most common manifestations of IgE-dependent cows’ milk protein hypersensitivity in infancy are skin changes with clinical manifestations of atopic dermatitis and regurgitation. In the case of non-IgE-mediated reactions, the gastrointestinal tract is most often involved, and the manifestation of such a course may be vomiting, blood in the stool, abdominal pain, lack of weight gain, and dysphagia [8,10]. In practice, both skin and gastrointestinal symptoms are extremely common in infants.

Prolonged exclusive breastfeeding with or without supplementation with hypoallergenic formulas for infants at risk has been recommended to prevent CMA. In the case of CMA, the current standard of care is strict avoidance of cow’s milk protein in the diet and the use of special formulas in non-breastfed individuals [11]. Formulas that can be considered effective for the dietary management of CMA include extensively hydrolyzed whey formula (EHWF), extensively hydrolyzed casein formula (EHCF), hydrolyzed rice formula, amino acid-based formula (AAF), or soy formula [12].

Cow milk-based extensively hydrolyzed formula (EHF) remains the recommended and preferred therapeutic choice, while AAF is reserved for the most severe cases [13]. The proper recommendation of AAF is a widely discussed topic due to its high cost, therefore limiting the abuse of the AAF base mixture and defining the group of patients who benefit most from such a treatment is very desirable.

We previously described the effect of AAF intervention in children with severe CMA in a real-life observational study [14]. Currently, using the same cohort, we aim to establish the phenotype of patients with CMA in whom the use of AAF will reveal the greatest clinical effect.

## 2. Materials and Methods

The details of our multicenter, real-life study have been described previously [14]. Briefly, it was conducted by 78 physicians with a group of 232 infants between 1 October 2021 and 1 March 2022. The study group consisted of children aged 0–12 months diagnosed by physicians with severe CMA (according to results of diagnostic procedures described in Appendix A (Appendix A). The criteria for inclusion into this study were as follows: (a) age 0–12 months, (b) severe CMA (severe clinical presentation, anaphylaxis, symptoms that have not fully resolved after using eHF, no improvement after elimination of CMP from maternal diet, and disorders of growth processes—decrease and/or lack of weight gain), and (c) indication for dietary management based on amino acid-based formula (Nutramigen PURAMINO). The exclusion criterion was the lack of indications for the use of AAF. Two study visits were scheduled. The time between the first and second visit was 5 weeks (±1 week) and depended on the doctor’s decision. At the first visit, written informed consent was obtained from all parents. The questionnaire completed by physicians during the first visit concerned the following: (i) diagnostic procedures used, (ii) the reason for AAF use, (iii) patient clinical characteristics, including sex, age, length, body weight, and family history (allergy occurrence in 1st-degree relatives), and (iv), the clinical picture and severity of CMA symptoms (skin, gastrointestinal, respiratory, and other symptoms shown in Appendix A (Appendix A). The questionnaire from the second visit concerned the severity of CMA symptoms and the subjective assessment of the response to treatment with AAF. The severity of clinical symptoms was assessed by the investigator on a three-point scale: 0—no symptoms (min), 1—mild, 2—moderate, and 3—severe symptoms (max).

In the present study, we conducted a post hoc analysis using clustering methods with validation in an independent cohort, followed by multivariate linear regression to achieve the assumed goals. Cluster analysis based on symptom severity at the 1st visit was performed. Clustering results were validated in the separate cohort with clinical characteristics given in Appendix A (Appendix A). The same profile of symptoms and the same severity score were implemented. The validation cohort included children with various severities of CMA: mild, moderate, and severe.

### Statistical Methods

Nominal variables were described with absolute and relative frequency. Numerical variables were described with mean and standard deviation in the case of normal distribution or median and interquartile range in other cases. Distribution normality was verified with the Shapiro–Wilk test as well as skewness and kurtosis. Variance homogeneity was verified with the Levene test. Cluster analysis based on symptoms severity was performed with a hierarchical method. V-fold cross-validation was applied to estimate the optimal number of clusters. As a result, four clusters were identified. Cluster D, which consisted of n = 3 patients, was excluded from further analysis due to low counts. For the analysis of clustering based on symptom severity in the validation sample, we used a model-based clustering as implemented in the R package BOS with the UMAP algorithm (as implemented in package uwot) to embed the posteriori probabilities in the two-dimensional space with the number of relevant neighbors set as 50 [15]. For numeric variables, including change in the severity of symptoms between visit II and visit I, comparisons between segments were performed with Anova analysis, Welch Anova analysis, or Kruskal–Wallis test, as appropriate. The post hoc evaluation was executed with the Tukey test and Dunn test with Bonferroni adjustment. Segment comparisons for categorical parameters were executed with Pearson’s Chi-square test. The significance of change in symptom severity from visit I to visit II was verified with a paired t-test (all symptoms variable) or Wilcoxon test (particular symptoms).

Linear regression analysis was performed in two stages. The dependent variable was the level of change in severity of all symptoms over time. Predictors were the available parameters other than the symptoms themselves. The first stage was performing univariate regression models on all predictors. The second stage was a multivariate linear regression model. Initial variable selection was based on the *p*-value from univariate models with the cut-off point of *p* = 0.157 [1]. Then, a stepwise approach was employed to select the final parameters for the multivariate model. The assessment of model fit included R2 and adjusted R2. Collinearity was verified with VIF indicators. Significance was indicated when alpha < 0.05. Data were analyzed with R statistical software, version 4.1.2.

## 3. Results

### 3.1. Baseline Characteristics and Cluster Analysis

The analyzed group consisted of 232 patients, out of which 137 (59.1%) were male. All patients participated in visit I and 226 patients participated in visit II. Six patients dropped out of this study due to missing their second visit. The baseline characteristics are given in Table 1.

There were 32 symptoms of CMA involved, grouped into skin, gastrointestinal, respiratory, and other symptoms. Based on the severity of the above symptoms, four clusters were identified.

Cluster A (GI, gastrointestinal) (n = 90, 38.8% of patients) consisted of children with moderate-to-severe gastrointestinal symptoms and mild other symptoms.

Cluster B (skin) (n = 79, 34.1% of patients) consisted of children with severe skin symptoms and mild other symptoms.

Cluster C (combined GI and skin) (n = 60, 25.9% of patients) consisted of children with moderate-to-severe gastrointestinal and severe skin symptoms and with mild other symptoms.

Cluster D (severe) (n = 3, 1.3% of patients) was composed of children with severe skin and gastrointestinal symptoms, and anxiety or sleep disorders.

Due to its size, cluster D was excluded from further analysis. Characteristics of the clusters are given in Figure 1. Between-cluster comparisons (Appendix A, Appendix A) confirmed significant differences in the severity of all skin symptoms except urticaria and in all gastrointestinal symptoms except food protein-induced enterocolitis syndrome (FPIES). The severity of other symptoms did not differ between clusters.

Age, height, and weight were significantly different between clusters A, B, and C (*p* = 0.038, *p* = 0.003, *p* = 0.003, respectively). All anthropometric parameters were significantly lower in children with moderate-to-severe gastrointestinal symptoms (cluster A) compared to children with severe skin symptoms (cluster B) (Table 2).

### 3.2. Validation of Clustering

The clustering of moderate-to-severe disease samples in a separate cohort revealed three clusters with the same pattern of symptom severity distribution across the three clusters as seen in the training dataset (Appendix A).

To test the results on the clustering of the data regardless of disease severity, we used the same clustering approach as described above. Two over-dispersed clusters in the testing dataset, regardless of the severity of the disease, were detected (Appendix A).

### 3.3. The Effect of the Intervention by Cluster

Differences in the severity scale of each symptom between visits I and II were assessed and compared within and between clusters (Table 3).

Within all clusters, significant clinical improvement was observed. Significant differences were observed between clusters with respect to the change in individual GI and skin symptom scores and the total symptom score (Table 3 and Figure 2).

Finally, a linear regression analysis was run for the change in the total severity scale (sum of symptoms with a severity higher than 0 on the first visit). All statistical predictors of the change in the total severity scale in univariate models are given in Table 4.

The multivariate model of linear regression showed that the severity score reduction after AAF treatment was significantly higher (i) in cluster C than in clusters A and cluster B (as cluster C served as the reference group in the model), (ii) in children with a positive family history of allergy, and (iii) in children with growth retardation at baseline (Table 5). The above model fit assessment resulted in an R2 of 0.449 and an adjusted R2 of 0.426.

## 4. Discussion

Currently, the selection of a modified formula in children with CMA is mainly based on the doctor’s experience. In clinical practice, amino acid-based formula is reserved for the nutritional management of infants diagnosed with CMA with a moderate-to-severe course, including those who failed to respond to a trial of EHF [16,17]. To the best of our knowledge, there have been no studies to date evaluating the clinical phenotype of severe CMA suggesting the best outcomes following the inclusion of AAF as first-line therapy. In the present study, we have shown for the first time that symptoms of severe cow’s milk protein allergy in infants are grouped into three distinct phenotypes—(i) a moderate-to severe gastrointestinal phenotype, (ii) a severe skin phenotype, and (iii) a combined moderate-to-severe gastrointestinal and severe skin phenotype. All these phenotypes have different responses to treatment with AAF. The most significant improvement after implementation of AAF was obtained in patients with combined phenotype with moderate-to-severe gastrointestinal and severe skin symptoms.

For all of those involved in taking care of children’s health, it is important to understand the multifaceted aspects of CMA, such as its clinical presentation, diagnosis, and dietary management, as well as its primary prevention. In the case of children with less severe CMA symptoms, such as mild-to-moderate cutaneous signs or vomiting, EHF is considered the first choice [16,17]. In a systematic review, Hill et al. concluded that patients with CMA who tolerated EHF did not achieve more benefits after introducing AAF, and in infants with a clinical presentation of colic, constipation, urticaria, eczema, or gastroesophageal reflux disease (GERD), feeding with EHF was sufficient to diminish the symptoms [18]. However, infants with CMA who had severe symptoms failed with EHF, and switching to AAF resulted in clinical improvements with weight gain, less regurgitations, improvements in transit stools, and relief of skin symptoms [18,19]. According to the ESPGHAN GI Committee practical guidelines, EAACI guidelines, and more recent research, AAF is the first line of treatment in infants and children with (i) anaphylaxis due to CMA, (ii) severe gastrointestinal symptoms, (iii) severe skin symptoms, (iv) growth impairment, or (v) multiple food allergies [13,18,20,21].

Our findings showed the most significant improvements following AAF treatment occurred in patients with a combined CMA phenotype with moderate-to-severe gastrointestinal and severe skin symptoms. These results are consistent with the statements of Venter et al. and Meyer et al., who concluded that patients who require an AAF often present with multisystem involvement and fall within the more severe spectrum of gastrointestinal allergies [22,23]. Similarly, catching up on height and weight was observed after the use of AAF in infants with severe non-IgE-mediated gastrointestinal and skin symptoms [18].

The biggest strength of the present study is the clustering of symptoms of severe CMA, independently of the CMA mechanism (IgE-mediated or non-IgE-mediated). This resulted in three distinct clinical phenotypes of severe CMA with variable responses to treatment with amino acid-based formula. Moreover, clustering has been successfully validated in a distinct population. It is likely that only two phenotypes, (i) gastrointestinal and (ii) skin, can be identified in CMA with a milder clinical presentation. This study also has some limitations. The diagnosis of CMA was based on clinical symptoms and a diagnostic elimination diet, and most patients did not have an oral food challenge (OFC). However, in severe CMA, delayed diagnosis has a harmful impact on the child’s health as allergen exposure results in life-threatening allergic reactions and an escalation of their underlying inflammatory status [5]. Moreover, an oral food challenge is contraindicated in infants with severe CMA due to the risk of anaphylaxis. Another weakness of our study is that the severity ratings were assigned based on the attending physician’s subjective judgment without standardized criteria, therefore the validity of these assessments may be questionable. However, we think that our approach is methodologically acceptable, and remains closest to the natural conditions of the everyday clinical practice of pediatricians. This was our goal in designing this real-life study. Furthermore, it would also be interesting to investigate the changes in the gut microbiome in the patients studied and correlate them with the improvements, and this will be carried out in a subsequent study.

## 5. Conclusions

In this real-life study of 232 infants with a severe allergy to milk protein, three distinct clusters of clinical presentation were identified: (i) gastrointestinal symptoms (A), (ii) skin symptoms (B), and (iii) combined gastrointestinal and skin symptoms (C). Between-cluster differences were observed in the age, height, and weight of children. Significant clinical improvement (lower symptoms severity score) after 5 weeks of amino acid-based formula intervention was observed in all clusters with the highest reduction in the total symptom severity score noted (i) in children with a combined gastrointestinal and skin phenotype, (ii) in children with a positive family history of allergy, and (iii) in children with growth retardation at baseline.

Phenotyping of CMA may facilitate the prediction of responses to the use of a specific milk replacer formula. This fact is of great clinical relevance, as the effectiveness of a specific elimination diet has implications for the correct diagnostic and therapeutic process in children with CMA.

## Figures and Tables

**Figure 1 nutrients-17-01809-f001:**
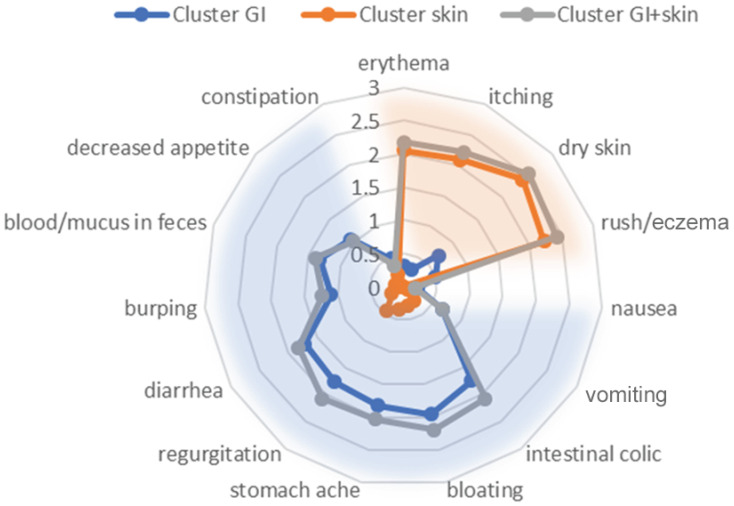
Radar graph of clusters’ differentiating symptoms.

**Figure 2 nutrients-17-01809-f002:**
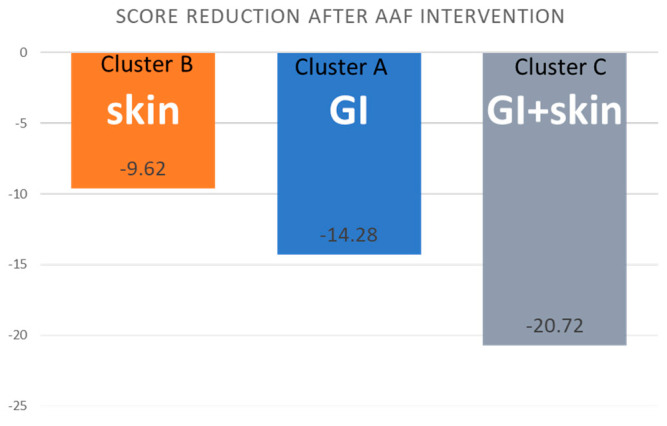
Reduction in total symptom scores (sum of all symptom severity scores) in defined clusters.

**Table 1 nutrients-17-01809-t001:** Baseline characteristics (n = 232).

Variable	n (% of Group)
Female	95 (40.9)
Male	137 (59.1)
Age, days * (mean ± SD)	142.02 ± 82.69
Age, months * (mean ± SD)	4.67 ± 2.72
Height, cm ** (mean ± SD)	64.13 ± 7.18
Weight, kg (mean ± SD)	6.53 ± 1.93
Height—percentile ***	
<3	3 (1.3)
3–10	27 (11.6)
10–25	57 (24.6)
25–50	58 (25.0)
50–75	52 (22.4)
75–90	25 (10.8)
90–97	7 (3.0)
>97	3 (1.3)
Weight—percentile ***	
<3	13 (5.6)
3–10	46 (19.8)
10–25	53 (22.8)
25–50	50 (21.6)
50–75	52 (22.4)
75–90	14 (6.0)
90–97	4 (1.7)
>97	0 (0.0)
Allergic diseases in family	156 (67.2)
mother	82 (35.3)
father	62 (26.7)
siblings	61 (26.3)

M—mean, SD—standard deviation, based on visit I. * Based on n = 230 due to missing data for two patients. ** Based on n = 231 due to missing data for one patient. *** Percentiles based on WHO growth references for part of patients and Polish Institute of Mother and Child growth references for others.

**Table 2 nutrients-17-01809-t002:** Age, height, and weight by cluster (n = 229).

Variable:	Cluster A (GI) n = 90	Cluster B (Skin) n = 79	Cluster C (GI + Skin) n = 60	*p*	Post Hoc Test ***
Age, days *	128.12 ± 85.31	160.48 ± 80.39	137.60 ± 79.15	0.038 ^1^	AB ^2^
Height, cm **	62.80 ± 8.45	65.49 ± 5.87	63.68 ± 5.72	0.003	AB
Weight, kg	6.15 ± 2.10	7.01 ± 1.91	6.50 ± 1.60	0.003	AB

SD—standard deviation, Significance of differences between groups verified with Anova analysis ^1^ and Kruskal–Wallis test. * Based on n = 230 due to missing data for two patients. ** Based on n = 231 230 due to missing data for one patient. *** Significantly different pairs of clusters, based on post hoc Tukey test ^2^ and Dunn test with Bonferroni adjustment. GI, gastrointestinal.

**Table 3 nutrients-17-01809-t003:** Change in symptom severity over time (before to after treatment) by cluster (n = 223).

Variables:	Cluster A (GI) n = 89	Cluster B (Skin) n = 76	Cluster C (GI + Skin) n = 58	*p*	Hoc Test *
**Total symptom scores (sum of all symptom severity scores) >0 on visit I**	**−14.28 ± 7.04**	**−9.62 ± 4.46**	**−20.72 ± 6.59**	<0.001 ^3^	AB, AC, BC ^2^
**Skin**					
Erythema	**−0.19 ± 0.67**	**−1.47 ± 0.89**	**−1.53 ± 0.80**	<0.001	AB, AC
Itching	**−0.19 ± 0.58**	**−1.59 ± 0.98**	**−1.67 ± 0.80**	<0.001	AB, AC
Dry skin	**−0.36 ± 0.77**	**−1.43 ± 0.91**	**−1.64 ± 0.77**	<0.001 ^1^	AB, AC ^2^
Rush/edema	**−0.37 ± 0.73**	**−1.75 ± 0.91**	**−1.91 ± 0.82**	<0.001	AB, AC
Urticaria	**−0.12 ± 0.52**	**−0.26 ± 0.64**	**−0.22 ± 0.65**	0.190	−
Angioedema	0.00 ± 0.15	−0.01 ± 0.50	−0.07 ± 0.37	0.420	−
**Gastrointestinal**					
Stomach ache **	**−1.51 ± 0.99**	−0.17 ± 0.82	**−1.50 ± 0.82**	<0.001	AB, BC
Diarrhea	**−1.54 ± 1.06**	**−0.17 ± 0.50**	**−1.66 ± 0.89**	<0.001	AB, BC
Nausea	**−0.15 ± 0.55**	0.00 ± 0.00	**−0.16 ± 0.41**	0.024	BC
Vomiting	**−0.62 ± 0.98**	−0.07 ± 0.30	**−0.60 ± 0.84**	<0.001	AB, BC
Regurgitation	**−1.26 ± 1.02**	**−0.24 ± 0.86**	**−1.57 ± 0.75**	<0.001	AB, BC
Burping	**−0.89 ± 1.02**	−0.01 ± 0.31	**−1.05 ± 0.98**	<0.001	AB, BC
Bloating	**−1.48 ± 0.99**	**−0.20 ± 0.71**	**−1.72 ± 0.79**	<0.001	AB, BC
Decreased appetite	**−0.93 ± 1.12**	0.00 ± 0.40	**−0.98 ± 1.02**	<0.001	AB, BC
Blood/mucus in feces	**−1.26 ± 1.12**	**−0.14 ± 0.45**	**−1.29 ± 0.94**	<0.001	AB, BC
Intestinal colic ***	**−1.43 ± 1.09**	**−0.18 ± 0.60**	**−1.83 ± 0.92**	<0.001	AB, BC, AC
Constipation	**−0.34 ± 0.77**	**−0.17 ± 0.62**	**−0.29 ± 0.75**	0.129	−
**Respiratory**					
Restricted nasal passage	−0.07 ± 0.47	**−0.14 ± 0.48**	−0.12 ± 0.65	0.517	−
Running nose	−0.06 ± 0.31	**−0.14 ± 0.56**	−0.10 ± 0.52	0.549	−
Chronic cough	−0.09 ± 0.42	−0.03 ± 0.16	−0.02 ± 0.35	0.677	−
Wheezing breath	**−0.12 ± 0.52**	**−0.13 ± 0.44**	−0.14 ± 0.63	0.801	−
Laryngeal edema	−0.02 ± 0.26	−0.03 ± 0.23	0.03 ± 0.32	0.549	−
Dyspnea	−0.10 ± 0.48	**−0.14 ± 0.51**	0.03 ± 0.32	0.068	−
**Other symptoms**					
Tearing eyes	0.00 ± 0.00	0.00 ± 0.16	0.00 ± 0.00	>0.999	−
Itching eyes	−0.01 ± 0.11	−0.01 ± 0.11	0.00 ± 0.00	0.697	−
Eye redness	0.00 ± 0.00	−0.04 ± 0.26	0.00 ± 0.00	0.143	−
Anxiety	**−0.13 ± 0.53**	−0.11 ± 0.42	−0.05 ± 0.39	0.434	−
Sleep disorders	−0.09 ± 0.47	−0.08 ± 0.36	0.00 ± 0.26	0.378	−
Apathy	−0.02 ± 0.21	−0.03 ± 0.16	−0.03 ± 0.26	0.777	−
Paleness	−0.02 ± 0.34	−0.08 ± 0.42	0.00 ± 0.00	0.177	−
Heavy sweating after a meal	−0.02 ± 0.21	0.00 ± 0.00	0.00 ± 0.00	0.471	−
Growing disorders	−0.13 ± 0.63	−0.08 ± 0.42	−0.03 ± 0.18	0.940	−

SD—standard deviation. Improvement of symptom severity calculated at visit II vs. visit I. Significance of differences between groups verified with Anova analysis ^1^, Welch Anova ^3^, and Kruskal–Wallis test. Statistical significance of within-group change in symptom severity was verified with paired *t* test (all symptoms > 0 on visit I) or Wilcoxon test (particular symptoms) and indicated with bolded font. * Significantly different pairs of clusters, based on post hoc Tukey test ^2^ and Dunn test with Bonferroni adjustment. GI, gastrointestinal. ** Infant restlessness/crying observed by caregivers during and after feeding was defined as “stomach ache”. *** Increased intestinal peristalsis identified by auscultation during a doctor’s visit in a child with the above symptoms was defined as “intestinal colic”.

**Table 4 nutrients-17-01809-t004:** Results of univariate regression for changes in symptom severity over time.

Variable	β	95% CI for β	Std. β	*p*
Cluster				
A (vs. B)	−4.67	−6.52 to −2.81	−0.62	<0.001
A (vs. C)	6.44	4.15 to 8.73	0.86	<0.001
B (vs. C)	11.11	9.21 to 13.00	1.49	<0.001
Sex, male	0.45	−1.56 to 2.45	0.06	0.660
Age at baseline days	0.01	0.00 to 0.02	0.13	0.058
Height at baseline, cm	0.17	0.03 to 0.31	0.16	0.020
Weight at baseline, kg	0.68	0.16 to 1.20	0.17	0.010
Allergic diseases in family	−1.73	−3.83 to 0.36	−0.23	0.104
Mother	−4.15	−6.15 to −2.14	−0.56	<0.001
Father	−2.64	−4.85 to −0.42	−0.35	0.020
Siblings	0.00	−2.23 to 2.23	0.00	0.997
Additional intervention *				
Antihistamine	−1.30	−3.92 to 1.32	−0.17	0.330
Emollients	−1.17	−3.17 to 0.83	−0.16	0.251
Probiotics	−3.05	−5.10 to −0.99	−0.41	0.004
Breastfeeding	0.58	−1.66 to 2.82	0.08	0.609
Reason for introducing AAF:				
Symptoms did not resolve after extensively hydrolyzed formula (EHF)	0.58	−1.74 to 2.90	0.08	0.623
Severe gastrointestinal symptoms	−4.67	−6.58 to −2.77	−0.63	<0.001
Severe atopic dermatitis	−1.73	−3.74 to 0.27	−0.23	0.090
Food allergy to multiple ingredients	−1.08	−5.29 to 3.14	−0.14	0.615
Lack of recovery after mother’s elimination diet	−1.59	−3.80 to 0.62	−0.21	0.157
Growth retardation	−5.72	−8.27 to −3.17	−0.77	<0.001
Anaphylaxis	−2.64	−17.43 to 12.14	−0.35	0.725
Other	−0.95	−5.51 to 3.61	−0.13	0.682

β—beta coefficient, CI—confidence interval, Std. β—standardized beta. For severity changes over time, only symptoms with severity > 0 in visit I were selected for each patient. * intervention added to amino acid formula (AAF). GI, gastrointestinal.

**Table 5 nutrients-17-01809-t005:** Results of multivariate linear regression for changes in symptom severity change over time.

Variable	β	95% CI for β	Std. β	*p*
Cluster				
A (GI)	4.18	2.02 to 6.33	−0.56	<0.001
B (skin)	9.73	7.70 to 11.75	0.74	<0.001
C (GI + skin) (reference)	-	-	-	-
Allergic diseases—mother	−2.14	−3.79 to −0.49	−0.29	0.011
Allergic diseases—father	−1.88	−3.62 to −0.15	−0.25	0.034
Reason for AAF implementation—growth retardation	−3.08	−5.16 to −1.00	−0.41	0.004

β—beta coefficient, CI—confidence interval, Std. β—standardized beta. For severity changes over time, only symptoms with severity > 0 in visit I were selected for each patient. AAF, amino acid-based formula; GI, gastrointestinal.

## Data Availability

The data that support the findings of this study are available on request from the corresponding author. The data are not publicly available due to privacy or ethical restrictions.

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
