# Peer review of "Clinical Phenotypes of Severe Cow’s Milk Protein Allergy with Various Responses to Amino Acid-Based Formula"

_nutrients, 2025, doi:10.3390/nu17111809_

Round 1

Reviewer 1 Report

Comments and Suggestions for Authors

The manuscript presents an interesting clustering analysis of infants with severe CMA. Several methodological issues must be addressed.

  1. The criteria used to define “severe CMA” are not described. Although the authors reference a previous study (Ref. 14), study is published in a non-English journal, which limits accessibility for an international audience. The diagnostic criteria for classifying CMA as "severe" is not also written. For the diagnosis of CMA, OFC must be performed. 
  2. The “indication for dietary management based on amino acid-based formula (AAF)” is described as a prerequisite for inclusion, yet the clinical criteria are not described.
  3. The severity scoring criteria for symptoms at the second visit are not described in sufficient detail. What constitutes “mild,” “moderate,” or “severe” for each symptom?
  4. The basis for symptom selection and their assignment in the clustering algorithm is not described. Were all 32 symptoms used equally in the clustering? Were they weighted or standardized in any way?

Author Response

The manuscript presents an interesting clustering analysis of infants with severe CMA. Several methodological issues must be addressed.

  1. The criteria used to define “severe CMA” are not described. Although the authors reference a previous study (Ref. 14), study is published in a non-English journal, which limits accessibility for an international audience. The diagnostic criteria for classifying CMA as "severe" is not also written. For the diagnosis of CMA, OFC must be performed. 

Res.: Thank the Reviewer for this question. Criteria were added: “…severe CMA (sever clinical presentation, anaphylaxis, symptoms that have not fully resolved after using eHF, no improvement after elimination of CMP from maternal diet, disorders of growth processes - decrease and/or lack of weight gain),…” Of course, we agree that for a confirmed diagnosis of food allergy, we need OFC. Diagnosis of severe milk allergy is a process that, after evaluation of clinical data, begins with an elimination diet test. Unfortunately, the re-exposure test in OFC is often difficult to implement in real-life conditions for children with severe milk allergy. Our study is of the real-life type and evaluates the effect of the elimination diet in children with suspected severe milk allergy (over 75% of included children did not have full improvement after EHF).

  1. The “indication for dietary management based on amino acid-based formula (AAF)” is described as a prerequisite for inclusion, yet the clinical criteria are not described.

Res.: As suggested, a sentence of explanation was added specifying the criteria for severe allergy to cow's milk protein, which was also an indication for intervention with AAF. It was a real-life study, so the doctor made the decision on AAF and included the patient in the study each time

  1. The severity scoring criteria for symptoms at the second visit are not described in sufficient detail. What constitutes “mild,” “moderate,” or “severe” for each symptom?

Res.: The severity of clinical symptoms was assessed by the investigator on a three-point scale: 0 - no symptoms (min), 1 - mild, 2 - moderate, and 3 – severe symptoms (max). This sentence of explanation is in the results section. This assessment was performed in the same way at both visits

  1. The basis for symptom selection and their assignment in the clustering algorithm is not described. Were all 32 symptoms used equally in the clustering? Were they weighted or standardized in any way?

Res.: All 32 symptoms recorded at baseline were included in the clustering algorithm, each contributing equally to the analysis. Standardization was not applied, as all symptoms were measured on the same ordinal scale (0–3), ensuring consistent measurement units across variables.

Thank you for your time, accurate and favorable reviews of our study and manuscript.

Reviewer 2 Report

Comments and Suggestions for Authors

The manuscript Clinical phenotypes of severe cow's milk protein allergy with 
various responses to amino acid-based formula is a follow-up study of a previous study from Majak's group with the same cohort of patients. They provide a rationale for the use of AAF on different groups of CMA patients classified according to their symptoms. The study is of interest. However, some aspects need to be addressed.

Line 49- Format citation.

Lines 64-68- Please explain why the choices of the different dietary approaches.

Lines 93 & 237- Please consider changing "presented" to "present" study.

Lines 105-106- Is this a supervised or non-supervised clustering analysis?

Figure 1 legend. Please revise English for correctness. I do not understand what Figure 1B reflects. Reduction in the second visit after treatment with the AAF diet? If so, this should come later. Please, improve the figure legends and the text for a better description of the results.

Sections 3.1 and 3.2. I think the results of these 2 sections should be merged and represented in an appropriate way (a graphical figure) to improve the readability and understanding of the results. I would not include part of them in the Supplementary material. For instance, the age of the different clusters is a relevant dataset that should be included in the results section.

Line 192. Please revise and correct.

Lines 191-195- In my opinion, it is difficult to infer the information contained in the Table to what is written in the text. Why do you affirm that "Multivariate model of linear regression showed that severity score reduction after AAF treatment was significantly higher: (i) in cluster C than in clusters A and cluster B", when cluster C is the reference cluster for comparisons?

Lines 200-211- Please revise abbreviations already used along the text (AAF).

Comments on the Quality of English Language

The use of English is mostly fine. There are some typos or minimal spelling/ writting mistakes that need to be addressed (see specific comments).

Author Response

The manuscript Clinical phenotypes of severe cow's milk protein allergy with 
various responses to amino acid-based formula is a follow-up study of a previous study from Majak's group with the same cohort of patients. They provide a rationale for the use of AAF on different groups of CMA patients classified according to their symptoms. The study is of interest. However, some aspects need to be addressed.

Line 49- Format citation.

Res.: Incorrect format has been unified

Lines 64-68- Please explain why the choices of the different dietary approaches.

Res.: According current DRACMA guidelines  the choice of milk formula during elimination depends on the clinical situation. The use of AAF is recommended in severe cases and in the absence of an adequate response to EHF. This sentence was added to the methods section

Lines 93 & 237- Please consider changing "presented" to "present" study.

Res.: Changes were made according to The Reviewer suggestion

Lines 105-106- Is this a supervised or non-supervised clustering analysis?

Res.: The clustering analysis was unsupervised. No outcome or target variable was used to guide the clustering process; clusters were formed solely based on symptom profiles.

Figure 1 legend. Please revise English for correctness. I do not understand what Figure 1B reflects. Reduction in the second visit after treatment with the AAF diet? If so, this should come later. Please, improve the figure legends and the text for a better description of the results.

Res.: As suggested, the engraving has been ungrouped, part B becoming a separate engraving with its own corrected legend. Thank the Reviewer for this comment, it makes it easier to interpret the results.

Sections 3.1 and 3.2. I think the results of these 2 sections should be merged and represented in an appropriate way (a graphical figure) to improve the readability and understanding of the results. I would not include part of them in the Supplementary material. For instance, the age of the different clusters is a relevant dataset that should be included in the results section.

Res.: As suggested, Sections 3.1 and 3.2. have been merged. Table 2 with important clinical data has been moved from the supplementary materials to the main manuscript. We agree that this change improves readability.

Line 192. Please revise and correct.

Res.: The sentence has been corrected according to the suggestions below

Lines 191-195- In my opinion, it is difficult to infer the information contained in the Table to what is written in the text. Why do you affirm that "Multivariate model of linear regression showed that severity score reduction after AAF treatment was significantly higher: (i) in cluster C than in clusters A and cluster B", when cluster C is the reference cluster for comparisons?

Res.: Thank the Reviewer for pointing this out. We agree that the sentence could be misinterpreted. In the multivariate regression model, cluster C was used as the reference category. The positive and statistically significant beta coefficients for clusters A and B (relative to cluster C) indicate that the reduction in symptom severity after AAF treatment was significantly weaker in clusters A and B compared to cluster C. Thus, symptom improvement was significantly greater in cluster C than in clusters A and B.

Lines 200-211- Please revise abbreviations already used along the text (AAF).

Res.: The paragraph has been corrected as suggested.

Thank you for your time, for your precise, thorough and favorable review of our work!

Round 2

Reviewer 1 Report

Comments and Suggestions for Authors

The authors appear to have addressed some of the reviewer comments to a certain extent. However, the response to the third comment—regarding the criteria for evaluating symptom severity—remains insufficient.

Stating that symptoms were assessed using a three-point scale does not adequately answer the concern. The authors must clearly define which clinical features were categorized as mild, moderate, or severe. If the severity ratings were assigned solely based on the attending physician’s subjective judgment, without standardized criteria, the validity of these assessments is questionable. Moreover, there are lacks of clarity on key clinical symptoms. For instance, can infants reliably report "stomachache"? What precisely do the authors mean by "intestinal colic" in this context? These terms require more precise definition to ensure interpretability and reproducibility of the findings.

Author Response

The authors appear to have addressed some of the reviewer comments to a certain extent.

Resp. We thank the reviewer for this comment/

However, the response to the third comment—regarding the criteria for evaluating symptom severity—remains insufficient. Stating that symptoms were assessed using a three-point scale does not adequately answer the concern. The authors must clearly define which clinical features were categorized as mildmoderate, or severe. If the severity ratings were assigned solely based on the attending physician’s subjective judgment, without standardized criteria, the validity of these assessments is questionable. 

Resp. Attempts at objective and comparable assessment of cow’s milk protein allergy symptoms in the youngest patients remain a clinical challenge. Currently, the Cow’s Milk-related Symptom Score (CoMiSS) is a recommended, helpful tool, in which the assessment of skin and respiratory symptoms is based solely on subjective categorization of the severity of symptoms on a 3-point scale from mild to severe. All symptoms in our study were assessed in the same way. Our study was conducted before (CoMiSS) validation results were published. We think that our approach is methodologically similar to the above and acceptable, and also remains closest to the natural conditions of everyday clinical practice of pediatricians. This was our goal in designing this real-life study. We added this comment to the text of the manuscript.

Moreover, there are lacks of clarity on key clinical symptoms. For instance, can infants reliably report "stomachache"? What precisely do the authors mean by "intestinal colic" in this context? These terms require more precise definition to ensure interpretability and reproducibility of the findings.

Resp. Indeed, both “stomach ache” and “intestinal colic” require clarification. Infant restlessness/crying observed by caregivers during and after feeding was defined as “stomach ache”. Increased intestinal peristalsis identified by auscultation during a doctor's visit in a child with the above symptoms was defined as "intestinal colic. " We added the above definitions to the description of Table 3.

Round 3

Reviewer 1 Report

Comments and Suggestions for Authors

Although the evaluation methods and survey instruments are not clearly described, making it difficult to fully assess the validity of the revisions, the authors appear to have addressed the concerns to a reasonable extent.